# Analysis of *Cyperus esculentus*–Soil Dynamic Behavior during Rotary Tillage Based on Discrete Element Method

Zhuang Zhao [1] , Dongwei Wang [2,*], Shuqi Shang [2], Jialin Hou [1], Xiaoning He [2], Zenghui Gao [3], Nan Xu [1], Zengcun Chang [1], Peng Guo [1] and Xiaoshuai Zheng [2]

1 College of Mechanical and Electronic Engineering, Shandong Agricultural University, Taian 271018, China
2 College of Mechanical and Electrical Engineering, Qingdao Agricultural University, Qingdao 266109, China
3 College of Mechanical and Electrical Engineering, Xinjiang Agricultural University, Urumqi 830052, China
* Correspondence: 200701031@qau.edu.cn

**Abstract:** Considering the problems of low soil fragmentation rates, high working resistance, and high energy consumption in the harvesting process of *Cyperus esculentus* in China, a method of *Cyperus esculentus* harvesting based on counter-rotation digging is proposed. The mechanism of interaction between the rotary tillage blade and *Cyperus esculentus*–soil is systematically investigated, and the vertical and horizontal disturbance performance of the positive and counter-rotating harvesting methods on soil and *Cyperus esculentus* is compared and analyzed. The results of the experiment showed that the intensity of soil and *Cyperus esculentus* disturbance by counter-rotation increased by 166.67% and 297.78%, respectively, and the effective disturbance time of soil and *Cyperus esculentus* increased by 133.33% compared to that of positive rotation. The working depth and rotation speed of the rotary tillage blade were the most significant for soil and *Cyperus esculentus* disturbance intensity. The working depth increased from 150 mm to 170 mm, and the soil disturbance intensity increased by 17.91% and 21.37% for positive and counter-rotating operation, respectively, and the rotation speed of the rotary tillage blade increased from 270 rpm to 330 rpm, and the soil disturbance intensity increased by 28.85% and 35.29%, respectively. Compared with the positive rotation operation, the *Cyperus esculentus* counter-rotation soil fragmentation rate increased by 4.09%, the *Cyperus esculentus* damage rate decreased by 10.69%, and the buried fruit rate decreased by 7.38%. This paper helps to understand the interaction mechanism between the rototiller and *Cyperus esculentus*–soil and lays a theoretical foundation for the subsequent design and optimization of the *Cyperus esculentus* digging device.

**Keywords:** rotary tillage blade; counter-rotating operation; *Cyperus esculentus*–soil disturbance; EDEM

## 1. Introduction

*Cyperus esculentus* is a new economic crop that integrates oil, food, pasture, forage, and greenery, with wide adaptability, high oil content, and high nutritional value [1–3]. *Cyperus esculentus* is mainly harvested by rotary excavation in most areas of China. The operation mode is divided into forward and counter-rotation according to the rotation direction of the rotary cutter [4,5]. Moreover, according to relevant studies, counter-rotation excavation can effectively improve the soil cutting rate [6,7]. In order to improve the harvesting efficiency of *Cyperus esculentus*, analysis of the dynamic interaction of the rotary tillage blade on the soil and the *Cyperus esculentus* in the forward and reverse rotation operation mode can lay the foundation for the optimization and design of the structural parameters and working parameters of the *Cyperus esculentus* digging device [8–11].

Identifying the mechanism of rotary tillage blade operating with *Cyperus esculentus* agglomerates can further improve *Cyperus esculentus* harvesting efficiency [11,12]. Due to the influence of soil properties and the complexity of *Cyperus esculentus*–soil movement during harvesting, traditional experimental methods cannot effectively measure the root

system and soil movement patterns of *Cyperus esculentus*. The discrete element method can assess the interaction mechanism between tillage tools and soil and reflect the dynamic changes of soil and root system during the simulation [13–18]. Scanlan and Davies used soil tracers and digital image analysis to analyze soil movement under four tillage practices, and the experiment showed that the vertical movement of soil varied between treatments [19]. Lenaerts et al. constructed segmented bendable straw particles in a DE-Meter++ simulation environment and calibrated their physical properties to realistic straw characteristics [20]. Songül Gürsoy et al. measured and simulated the soil displacement capacity and cutting forces using discrete element simulation tests [21], which showed that the displacement was most significant around the center of the machine path and decreased farther away from the center. Saunders et al. used the discrete element method to simulate the effects of forward plow speed, working depth, and the working angle on force and soil movement, and the results showed that EDEM could accurately simulate tillage force and soil movement [22]. Du et al. analyzed the influence law of rotary tillage knives and helical horizontal knives on soil disturbance capacity under different working conditions by combining a discrete element simulation test system [23]. The results showed that the tillage depth had the most significant influence on tillage characteristics. Fang et al. used the tracer block method to measure soil displacement in soil tank experiments. They analyzed the soil displacement and movement mechanism in experimental and simulation data [24]. The results showed that the movement displacement of the shallow soil particles was the largest, followed by the middle soil, and the smallest in the deep soil.

The harvesting performance of the *Cyperus esculentus* digging device is mainly determined by the structural parameters and working parameters, such as rotary tiller structural parameters, forward speed, rotation speed, and working depth and other working parameters. Currently, domestic and foreign scholars apply the bionic principle to the design work of structural parameters of tillage components. They further improve the tillage efficiency by changing the structural parameters such as the radius of rotation of the rotary tillage blade, the inclination of shovel blade, and the shape of the edge curve [25,26]. Due to the complexity of soil properties, the current research related to *Cyperus esculentus* excavation devices mainly focuses on the analysis of mechanical properties and lacks an analysis of the root system and soil movement behavior of *Cyperus esculentus* [27,28].

The purpose of this study is to clarify the root–soil disturbance and movement law of *Cyperus esculentus* under the positive and counter-rotating operation mode and establish a soil–*Cyperus esculentus*–rotary tillage blade discrete element model. The influence of the forward speed, rotation speed, and working depth of the rotary tillage blade on the harvesting efficiency and torque of the blade shaft of the *Cyperus esculentus* was investigated, which provides a basis for decisions on the optimization and design of the subsequent *Cyperus esculentus* digging device.

## 2. Materials and Methods

### 2.1. Machine Configuration

In order to compare and analyze the action law of the rotary tillage blade on the root soil system of *Cyperus esculentus* in the positive and counter-rotation state, two *Cyperus esculentus* digging devices were selected in the EDEM simulation experiment. One type is the counter-rotation *Cyperus esculentus* digging device and the other is the positive rotation *Cyperus esculentus* digging device. The counter-rotating *Cyperus esculentus* digging device is shown in Figure 1. The rotary tillage blade of the counter-rotating operation is used counterclockwise to harvest *Cyperus esculentus* and it consists of a soil-directed surface, a universal joint, a three-point suspension, a gearbox, a drive shaft, a counter-rotating rotary tillage blade, chains, and left- and right-side baffles. The positive rotating *Cyperus esculentus* digging device is shown in Figure 2; the rotary tillage blade of the positive rotating operation is used clockwise to harvest *Cyperus esculentus*.

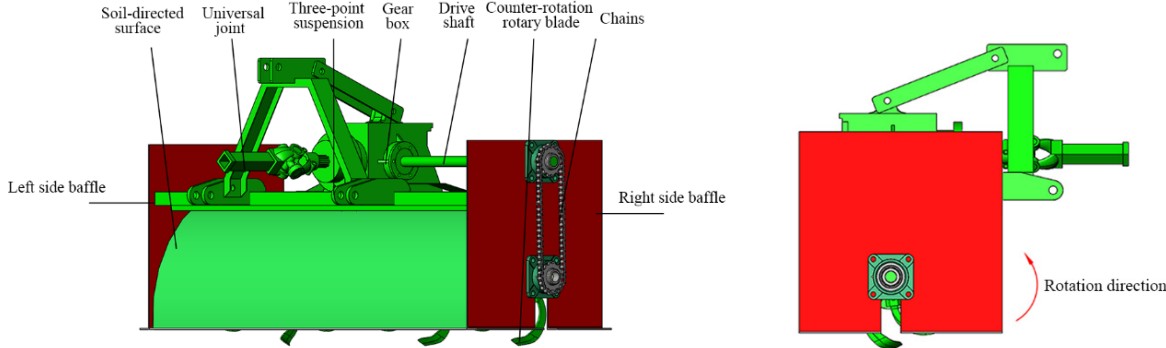

**Figure 1.** The counter-rotating *Cyperus esculentus* digging device.

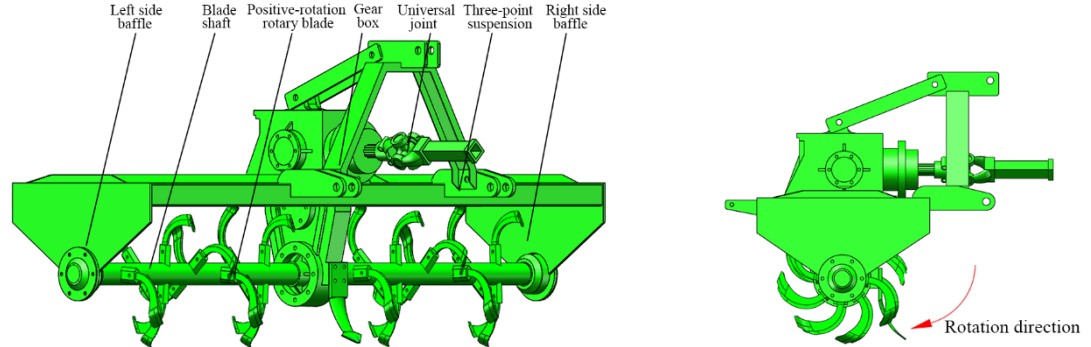

**Figure 2.** The positive-rotating *Cyperus esculentus* digging device.

To ensure the accuracy of the simulation and field trials, the construction and instal­lation parameters of the rotary cutter are the same for both positive and counter-rotating digging devices. In this paper, the rotary tillage blade is arranged in a double-spiral manner on the counter-rotating *Cyperus esculentus* digging devices, and the phase angle of the rotary tillage blade is 45°. Combined with the *Cyperus esculentus* planting pattern, the rotary tillage blade installation spacing is set to 130 mm, and the structural parameters of the rotary tillage blade are shown in Figure 3.

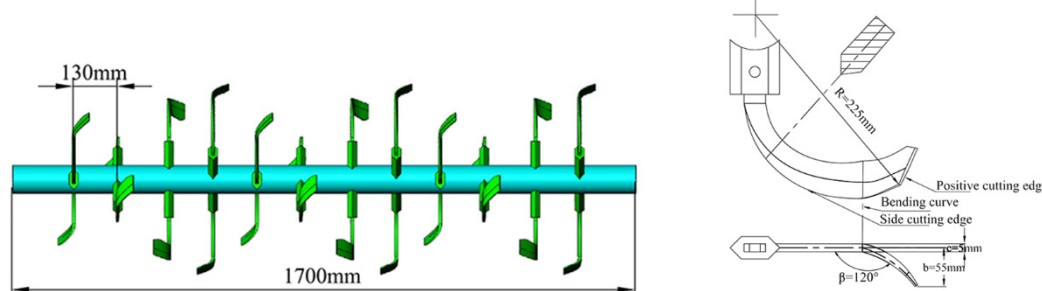

**Figure 3.** The Structure sketch of rotary tillage blade.

### 2.2. Principle of Positive and Counter-Rotating Harvesting of Cyperus esculentus

For counter-rotation operation, the rotation direction of the blade shaft is opposite to the rotation direction of the driving wheel of the machine, and the rotary tillage blade transitions from the operated area to the unoperated area and starts to cut the soil from the bottom to the ground. Due to the guiding effect of the retainer plate, the *Cyperus esculentus* agglomerates are transported backward along the tangential direction of the retainer plate, and the mechanism of counter-rotation operation of the rotary tillage blade is shown in

Figure 4a. For positive rotation operation, the rotation direction of the blade shaft is the same as the rotation direction of the driving wheel of the machine during the positive rotation operation, and the rotary tillage blade transitions from the unoperated area to the operated area and starts to cut the soil from top to bottom. The mechanism of rotary cutter positive rotation is shown in Figure 4b.

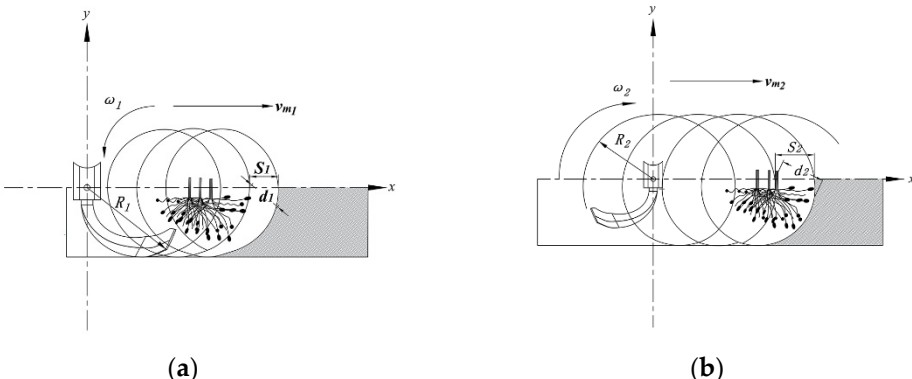

(**a**)  (**b**)

**Figure 4.** Harvesting mechanisms of positive-rotating and counter-rotating rotation. (**a**) Harvesting mechanisms of counter-rotation operation, where $v_{m1}$ is machine advance speed of counter-rotation, m/s; $\omega_1$ is tool shaft rotation speed of counter-rotation, rad/s; $S_1$ is soil cutting pitch of counter-rotation, mm; $d_1$ is counter-rotating movement of the soil to the cutting thickness, mm; $R_1$ is motion radius of gyration of counter-rotation, mm. (**b**) Harvesting mechanisms of positive-rotating, where $v_{m2}$ is machine advance speed of positive-rotating, m/s; $\omega_2$ is tool shaft rotation speed of positive-rotating, rad/s; $S_2$ is soil cutting pitch of positive-rotating, mm; $d_2$ is positive-rotating movement of the soil to the cutting thickness, mm; $R_2$ is motion radius of gyration of counter-rotation, mm.

The rotary tillage blade cutting speed is the main factor affecting the fragmentation of *Cyperus esculentus* agglomerates. Combined with the rotary tillage blade under positive and counter-rotation operation for the *Cyperus esculentus* cutting process, the rotary tillage blade tip cutting speed was obtained [29]:

$$\begin{cases} v_1 = v_{m1}\sqrt{1 + \lambda^2 - 2\lambda \sin \omega_1 t} \\ v_2 = v_{m2}\sqrt{1 + \lambda^2 + 2\lambda \sin \omega_2 t} \end{cases} \tag{1}$$

where $v_1$ is positive-rotating cutting soil speed, m/s; $v_2$ is counter-rotating cutting soil speed, m/s; $\lambda$ is rotary speed ratio; $t$ is time, s.

The thickness of soil cutting is the main factor affecting the fragmentation of *Cyperus esculentus* agglomerates. Combined with the rotary tillage blade under positive and counter-rotation operation for the *Cyperus esculentus* cutting process, the thickness of soil cutting was obtained [29]:

$$\begin{cases} d_1 = \frac{2\pi v_{m1}}{z_1 \omega_1} \sin\left(\frac{\lambda \cos \omega_1 t}{\sqrt{\lambda^2 - 2\lambda \sin \omega_1 t + 1}}\right) \\ d_2 = -\frac{2\pi v_{m2}}{z_2 \omega_2} \sin\left(\frac{\lambda \cos \omega_2 t}{\sqrt{\lambda^2 + 2\lambda \sin \omega_1 t + 1}}\right) \end{cases} \tag{2}$$

where $z_1$ is the number of blades in the same plane of positive-rotating; $z_2$ is the number of blades in the same plane of counter-rotating.

Investigating the operation mechanism of a rotary tillage blade for *Cyperus esculentus* agglomerates in positive and counter-rotation mode, the main operating parameters affecting rotary tillage blade harvesting are forward speed, rotary tillage blade rotation speed, and working depth. In order to further analyze the operation law of the positive and counter-rotation operation mode for *Cyperus esculentus* agglomerates under different operating parameters, this paper combines a discrete element simulation test and a soil bin test for systematic research and analysis.

*2.3. EDEM Modeling*

2.3.1. Soil Physical Properties and Discrete Element Modeling

Soil physical characteristics tests were conducted at the *Cyperus esculentus* planting base in Minquan County, Henan Province. According to the underground distribution state of *Cyperus esculentus*, the soil was divided into four areas, i.e., 0~50 mm, 50~100 mm, 100~150 mm, and 150~200 mm for soil test analysis.

The soil particle size test was carried out using a standard inspection sieve of different pore sizes (GB/T6003.1), and the test soil was poured into the uppermost standard sieve and sieved at all levels through standard sieves of different pore sizes. Finally, the soil from each sieve was collected and weighed with an electronic scale (accuracy of 0.01 g). The test results showed that the proportion of soil particle sizes ranging from 1.5 to 0.1 mm was 16.73%, 0.1 to 0.075 mm was 21.97%, 0.075 to 0.05 mm was 29.54%, and ≤0.05 mm was 31.76%.

In this paper, a soil cutting ring knife with a volume V of 100 cm$^3$ was used to take several samples in the soil areas of 0–50 mm, 50–100 mm, 100–150 mm, and 150–200 mm. The mass was measured using an electronic scale to record the total mass $m_1$ as well as the mass of the ring knife $m_0$, respectively, and the soil density was the ratio of the measured soil mass to the volume, calculated as shown in Equation (4). The test results are shown in Table 1.

$$\rho = \frac{m_1 - m_0}{V} \tag{3}$$

where $\rho$ is Soil density, g/cm$^3$; $m_1$ is total mass of ring knife and soil, g; $m_0$ is Mass of the ring knife, g; $V$ is volume of soil, cm$^3$.

**Table 1.** Soil physical parameters.

| Parameters | Depth of Soil | | | | | | | |
|---|---|---|---|---|---|---|---|---|
| | 0~50 mm | | 50~100 mm | | 100~150 mm | | 150~200 mm | |
| | Field | EDEM | Field | EDEM | Field | EDEM | Field | EDEM |
| Density (kg/m$^3$) | 1596 | 1596 | 1654 | 1654 | 1736 | 1736 | 1787 | 1787 |
| Moisture content (%) | 6.5 | / | 7.8 | / | 8.5 | / | 10.5 | / |

The soil moisture content was influenced by the degree of disturbance of *Cyperus esculentus*–soil. The soil moisture content test was carried out using aluminum boxes of mass $m_2$ for soil sampling in different soil layers and weighed using an electronic balance with a total mass of $m_3$. The soil samples were put into the soil and a 101-3A electric blast dryer was used for drying until the mass was constant and the mass was recorded as $m_4$. The calculation formula is shown in equation (4), and the test results are shown in Table 1.

$$H_t = \frac{m_3 - m_4}{m_3 - m_2} \times 100 \tag{4}$$

where $H_t$ is soil moisture content, %; $m_2$ is the aluminum box mass, g; $m_3$ is the aluminum box with wet soil mass, g; $m_4$ is the aluminum box with dry soil mass, g.

In recent years, many scholars have used discrete elements and finite elements for soil environment simulation to study the operating mechanism of soil and tillage components in depth [30–33]. The discrete element method treats the soil as consisting of particles with gaps, allowing the soil particles to move, rotate, and deform, while the soil particle gaps can be compressed, separated, or slid [34,35]. Ucgul et al. conducted a comparative test of particle models with 5 mm and 10 mm radii for EDEM soil particles. The test results showed that the most accurate results were obtained when using smaller particles, but this significantly increased the computational time [36]. To further improve the simulation efficiency, this paper treats the soil as a discontinuous discrete medium, sets the soil radius as 2.5 mm, and also simplifies the discrete element modeling of soil particles by dividing the

soil model into the following three types: single-grain spherical model, two-grain spherical model, and lumpy particle model, as shown in Figure 5.

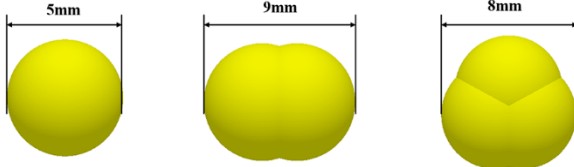

**Figure 5.** Soil discrete element particle model.

### 2.3.2. *Cyperus esculentus* Discrete Element Modeling

In order to further observe the motion trajectory of *Cyperus esculentus* during the simulation, a rigid modeling method was used to reorganize the *Cyperus esculentus* model. In order to improve the accuracy of the simulation test and accurately simulate the contact relationship between *Cyperus esculentus* and soil, the actual physical characteristics of *Cyperus esculentus* were combined to simplify the modeling of *Cyperus esculentus*. The friction and entanglement between the *Cyperus esculentus* root whiskers in the process of root modeling were ignored, and the primary root system was constructed to form a discrete element model of the rigid body *Cyperus esculentus*, as shown in Figure 6.

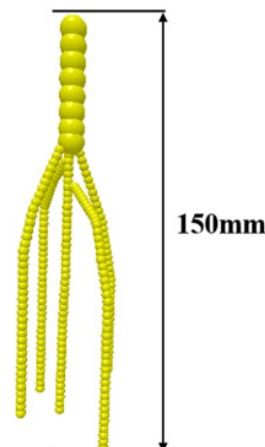

**Figure 6.** Discrete element model for the *Cyperus esculentus*.

### 2.3.3. Discrete Element Model

The soil particle modeling work in this study was performed using EDEM2020. The modeling of the contact between two particles consisted mainly of springs, damping, and friction [37], as shown in Figure 7. The interaction between two particles was modeled by a linear spring Kn with a constant in the normal direction and a linear spring with a constant Kt in the tangential direction in series with a friction coefficient u [38]. The equation of the contact force between the particles is as follows [37,38].

$$F_{n1} = \frac{4}{3} E^* \sqrt{R^*} \delta_n^{\frac{3}{2}} \tag{5}$$

$$\begin{cases} F_{t1} = -S_t \delta_t \\ \delta_t = 8G^* \sqrt{R^* \delta_t} \end{cases} \tag{6}$$

where $F_{n1}$ is the normal force between particles, N; $E^*$ is the equivalent Young's modulus, Pa; $R^*$ is the soil particle model equivalence radius, m; $\delta_n$ is the amount of normal overlap, m; $F_{t1}$ is the tangential force between particles, N; $G^*$ is the equivalent tangential modulus, Pa; $S_t$ is the tangential stiffness, N/m; $\delta_t$ is the amount of tangential overlap, m.

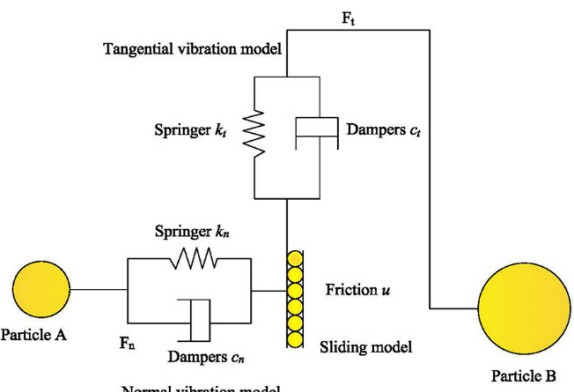

**Figure 7.** Discrete element particle model contact principle.

To further observe the dynamic interaction relationship between the rotary cutter and the soil, Hertz–Mindlin with bonding is used as a soil particle contact model in this paper, which can bond two adjacent soil particles together by bonding force while the bonding force between the soils can withstand tangential and normal displacements [39,40], as shown in Figure 8. During the actual field operation, there are adhesion forces between soil and soil, and there is a forced relationship between the rototiller and soil, and the model can simulate the bonding action between soil particles and the phenomenon of soil particle fragmentation.

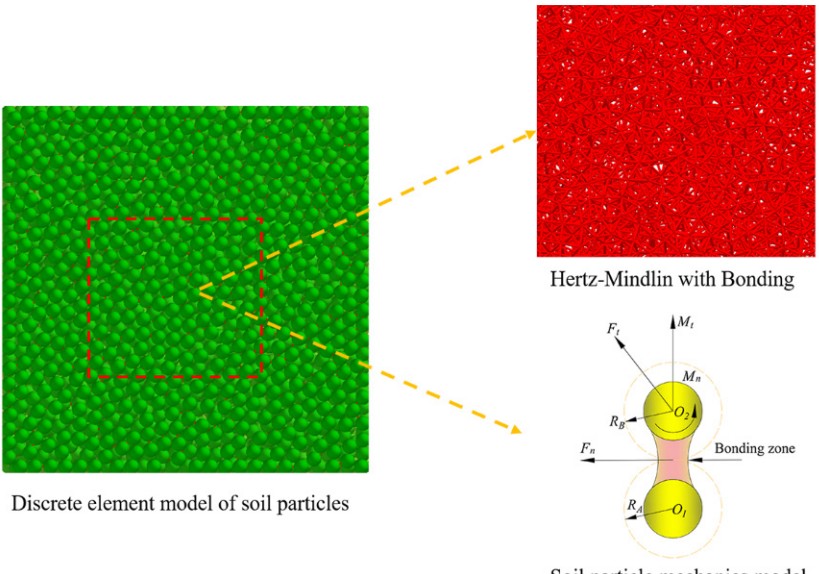

**Figure 8.** Soil particle contact model. Where O1 and O2 is the center of the circle of the soil model; RA and RB is the radius of the soil model, $m$; $F_n$ is normal contact force, N; $F_t$ is tangential contact force, N; $M_n$ is normal moment, $N_m$; $M_t$ is tangential moment, $N_m$.

When the bonding is formed between soils, the following contact relationships exist between soil particles [41,42]:

$$\begin{cases} \delta F_n = -v_n k_n A \delta_t \\ \delta F_t = -v_t k_t A \delta_t \\ \delta T_n = -\omega_n k_t A \delta_t \\ \delta T_t = -\omega_t k_t \frac{j}{2} \delta_t \end{cases} \tag{7}$$

where $V_n$ is normal velocity, m/s; $V_t$ is tangential velocity, m/s; $k_n$ is normal stiffness, N/m; $k_t$ is tangential stiffness, N/m; $A$ is unit contact area, mm$^2$; $J$ is moment of inertia, mm$^4$; $\delta_t$ is time step, s.

The bond between soil particles will break when the normal and tangential stresses reach specific extreme values, and when the bond breaks, the following contact relations exist [41,42]:

$$\begin{cases} \sigma_{\max} < \frac{-F_n}{A} + \frac{2M_t}{J} R_B \\ \tau_{\max} < \frac{-F_t}{A} + \frac{M_n}{J} R_B \end{cases} \tag{8}$$

where $\sigma_{max}$ is normal stress, N; $\tau_{max}$ is tangential stress, N; $R_B$ is radius of particle bonding, mm.

### 2.3.4. Discrete Element Simulation Parameters

The discrete element particle material properties and contact parameters between particles are closely related to the establishment of the discrete element test model. In this paper, to improve the accuracy and efficiency of the discrete element simulation test, the parameters required for the simulation test were determined by combining the previous paper for the determination of the fundamental physical parameters of the soil and by referring to the relevant literature [43,44], as shown in Table 2.

**Table 2.** Simulation parameters used in the discrete element method.

| Parameters | Depth of Soil | | | |
| --- | --- | --- | --- | --- |
| | 0~50 mm | 50~100 mm | 100~150 mm | 150~200 mm |
| Density of soil particles (kg/m$^3$) | 1596 | 1645 | 1736 | 1787 |
| Poisson's ratio of soil | 0.3 | 0.3 | 0.3 | 0.3 |
| Shear modulus of sand (Pa) | $1.0 \times 10^6$ | $1.0 \times 10^6$ | $1.0 \times 10^6$ | $1.0 \times 10^6$ |
| Coefficient of restitution, soil–soil | 0.33 | 0.38 | 0.38 | 0.41 |
| Coefficient of static friction, soil–soil | 0.42 | 0.45 | 0.45 | 0.47 |
| Coefficient of rolling friction, soil–soil | 0.35 | 0.37 | 0.37 | 0.39 |
| Density of steel (kg/m$^3$) | 7865 | 7865 | 7865 | 7865 |
| Poisson's ratio of steel | 0.3 | 0.3 | 0.3 | 0.3 |
| Shear modulus of steel (Pa) | $7.9 \times 10^{10}$ | $7.9 \times 10^{10}$ | $7.9 \times 10^{10}$ | $7.9 \times 10^{10}$ |
| Coefficient of restitution, soil–steel | 0.21 | 0.23 | 0.23 | 0.25 |
| Coefficient of static friction, soil–steel | 0.45 | 0.46 | 0.46 | 0.48 |
| Coefficient of rolling friction, soil–steel | 0.55 | 0.57 | 0.57 | 0.59 |
| Density of root (kg/m$^3$) | 1070 | 1070 | 1070 | 1070 |
| Poisson's ratio of root | 0.56 | 0.56 | 0.56 | 0.56 |
| Shear modulus of root (Pa) | $0.93 \times 10^5$ | $0.93 \times 10^5$ | $0.93 \times 10^5$ | $0.93 \times 10^5$ |
| Coefficient of restitution, root–steel | 0.42 | 0.43 | 0.43 | 0.45 |
| Coefficient of static friction, root–steel | 0.13 | 0.14 | 0.14 | 0.17 |
| Coefficient of rolling friction, root–steel | 0.04 | 0.05 | 0.05 | 0.07 |
| Coefficient of restitution, root–soil | 0.14 | 0.15 | 0.15 | 0.16 |
| Coefficient of static friction, root–soil | 0.42 | 0.44 | 0.44 | 0.45 |
| Coefficient of rolling friction, root–soil | 0.05 | 0.06 | 0.06 | 0.08 |

### 2.3.5. Discrete Element Model for *Cyperus esculentus* Agglomerates

In order to improve the efficiency and accuracy of the discrete element simulation test and to meet the actual growth state of the *Cyperus esculentus* plant, this paper establishes 1800 mm × 1200 mm × 200 mm (length × width × height). The discrete element simulation model was combined with the actual planting pattern of *Cyperus esculentus*. Two particle factories were set up to generate soil and *Cyperus esculentus* with an interval of 0.02 s. The parameters for the location of the *Cyperus esculentus* root system were set with a cubic position type, and the spacing of the *Cyperus esculentus* plant rows was set to 150 mm × 150 mm. The time the soil particles reached a stable state was 0.3 s and the simulation duration was 4 s. In reference to the relevant literature [2,6,12,35], the *Cyperus esculentus* planting depth

is 150~170 mm, harvesting speed is 0.5~0.9 m/s and the rotation speed of the rotary tillage blade is 270~330 rpm. To improve the accuracy of discrete element simulation tests the working depth of the rotary tillage blade is 150 mm, the forward speed is 0.5 m/s, and the rotation speed of the rotary tillage blade is 270 r/min.

In order to compare and analyze the harvesting performance of positive and counter-rotating *Cyperus esculentus* under different working parameters, two discrete meta-areas, A and B, were established and there are the same soil properties in areas A and B. The A area was divided into four parts vertically, namely Vertical Section 1, Vertical Section 2, Vertical Section 3, and Vertical Section 4, and the B area was divided into three sections horizontally, namely Horizontal Section 1, Horizontal Section 2, and Horizontal Section 3 to compare and analyze the *Cyperus esculentus*–soil disturbance in A and B areas during the positive and reverse rotation harvesting process. The discrete element model of *Cyperus esculentus* agglomerates is shown in Figure 9.

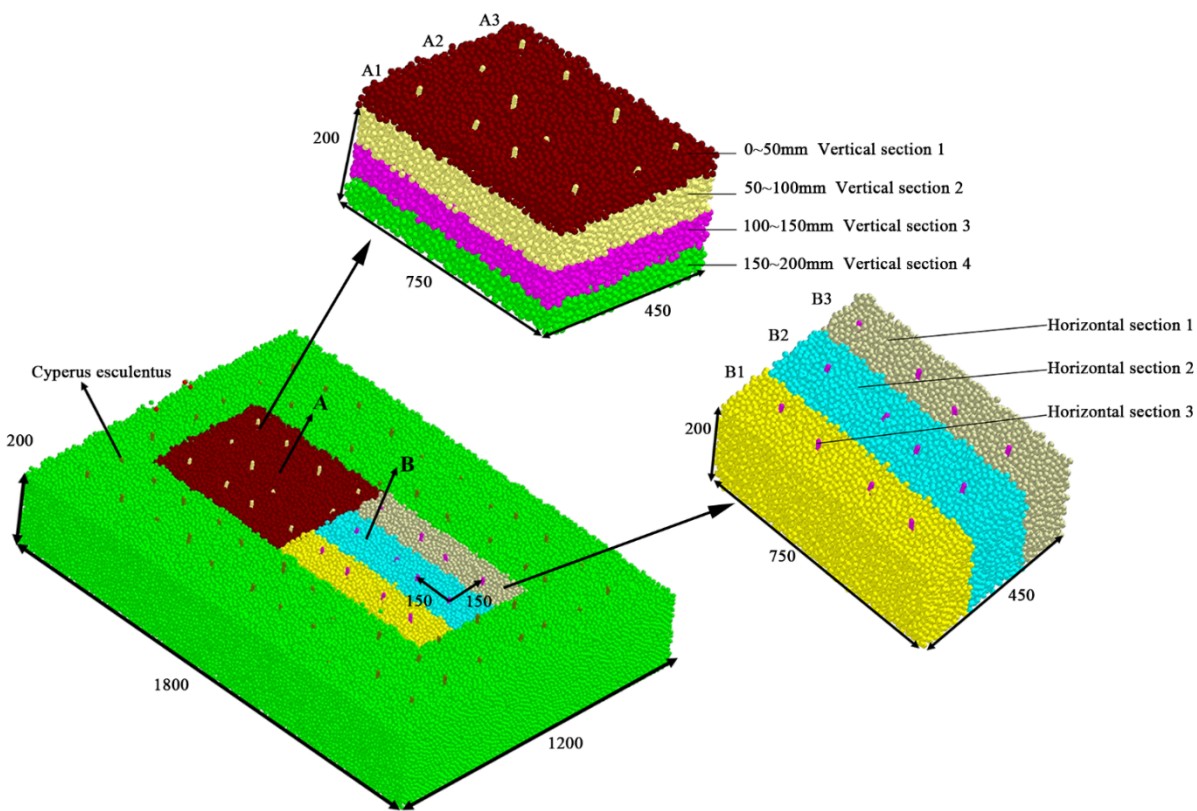

**Figure 9.** Discrete element model for *Cyperus esculentus* agglomerates.

### 2.4. Cyperus esculentus Field Harvesting Test
Test Method

To further validate the operational performance of the *Cyperus esculentus* counter-rotating digging device, *Cyperus esculentus* harvesting operation tests were conducted by combining theoretical analysis and simulation test results. The field harvesting test of *Cyperus esculentus* was conducted in December 2021 at the demonstration base of Qingdao Agricultural University, Qingdao, Shandong Province, China. In this paper, soil sampling within 0~200 mm was carried out using a ring knife, soil moisture content, density, and other parameters were measured for the sampled soil, and the test results are shown in Table 1. The analysis of the *Cyperus esculentus* planting pattern was also conducted, focusing on testing the *Cyperus esculentus* row spacing, planting depth, etc., as shown in Figure 10.

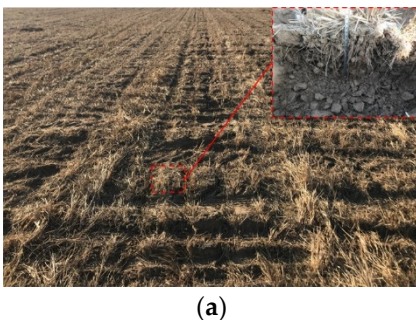 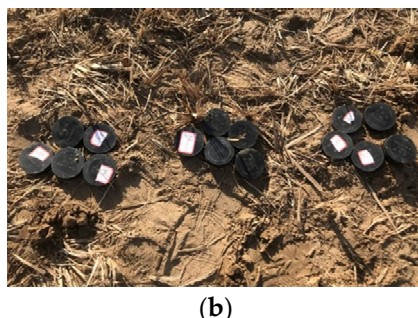

(**a**)          (**b**)

**Figure 10.** *Cyperus esculentus*-soil basic physical properties test. (**a**) Measurement of *Cyperus esculentus* planting pattern; (**b**) experimental soil sampling.

The main field trial instruments included a YDRZ-4L ring knife (Hangzhou Daji Photoelectric Instrument Ltd., Hangzhou, China), an LD-JS20 Soil hardness tester (Shandong Lainde Intelligent Technology Ltd., Shandong, China), a WKT-M1 soil moisture meter (Jiangsu Weikite Instruments Ltd.), WT-CF series high-precision electronic scales (Changzhou Wantai Balance Instruments Ltd., Changzhou, China, matching power: 60 hp), and a steel frame tape measure (Hunan Maojun Baogong Electronics Ltd., Changsha, China, range: 0~150 m, accuracy: 1 mm).

A planting area of 50 m in length and 2 m in width was selected for the field harvesting comparison test of *Cyperus esculentus*. To ensure the accuracy of the test, the working parameters of the counter-rotating digging device and the positive-rotating digging device were kept the same, the rotation speed of rotary tillage blade was 270 rpm, the forward speed was 0.5 m/s, and the working depth was 150 mm. With the soil fragmentation rate $Y_1$, the damage rate of *Cyperus esculentus* $Y_2$ and the buried fruit rate of *Cyperus esculentus* $Y_3$ were used as the test indexes for the analysis of the test results, and the test was repeated five times. The average value of the test results was taken, and the field operation of the *Cyperus esculentus* is shown in Figure 11.

$$Y_1 = \frac{m_a - m_b}{m_a} \times 100\% \tag{9}$$

$$Y_2 = \frac{m_d}{m_x + m_y + m_z} \times 100\% \tag{10}$$

$$Y_3 = \frac{m_y}{m_x + m_y + m_z} \times 100\% \tag{11}$$

where $m_a$ is the total mass of soil in the whole tillage layer in the test area, g; $m_b$ is the mass of the soil block with the longest side greater than 3 cm in the test area, g; $m_x$ is the mass of *Cyperus esculentus* tubers on the ground in the test area, g; $m_y$ is the mass of *Cyperus esculentus* tubers buried in the soil in the test area, g; $m_z$ is the mass of *Cyperus esculentus* tubers harvested from the test area, g; $m_d$ is the mass of damaged *Cyperus esculentus* tubers harvested from the test area, g.

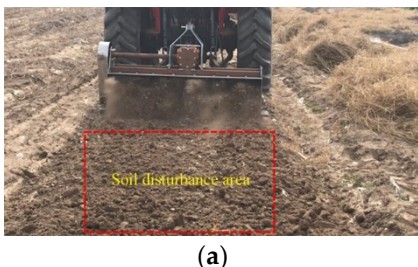 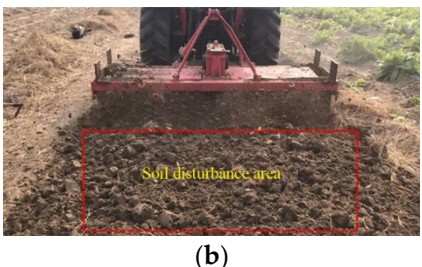

(**a**)          (**b**)

**Figure 11.** *Cyperus esculentus* field harvesting test. (**a**) Counter-rotating digging device; (**b**) positive-rotating digging device.

## 3. Results and Analysis

### 3.1. The Analysis of Cyperus esculentus–Soil Movement Behavior in Discrete Element Method

3.1.1. The Analysis of *Cyperus esculentus*–Soil Movement Behavior in Counter-Rotation Operation

To further improve the accuracy of the discrete element simulation test, a simplified model of the parametric mechanism of the *Cyperus esculentus* digging device was established with the premise of ensuring the primary function of the *Cyperus esculentus* device, combined with the basic principle of equivalent simplification of the mechanism, and the counter-rotating device mainly retained the rotary tillage blade shaft and the retaining plate for the discrete element simulation test. The movement process of *Cyperus esculentus* under the operation of a counter-rotating digging device is shown in Figure 12. In order to analyze the movement direction of soil particles *Cyperus esculentus*, the particle morphology was set as a vector in this paper. As shown in Figure 12 a, at 0.5 s, when the rotary tillage blade just touched the soil, the soil under the *Cyperus esculentus* agglomerate began to disturb, and with the compound movement of the rotary tillage blade, the *Cyperus esculentus* agglomerate was slightly disturbed by the soil and began to move along the tangential direction of the retaining plate. As shown in Figure 12b, with the continuous counter-rotation operation, the rotary tillage blade starts to contact the *Cyperus esculentus* agglomerates in area A. At this time, the rotary tillage blade moves from the bottom to the top, from far to near, to carry out the *Cyperus esculentus* counter-rotation throwing operation. As shown in Figure 12c,d, with the simulation test, the root system and soil of *Cyperus esculentus* under different soil layers in area A moved backward continuously and in an orderly way along the tangential direction of the retaining plate under the action of the rotary tillage blade, and when the rotary tillage blade operated in area B, the disturbance process and movement pattern of *Cyperus esculentus* roots system and soil was the same as that in area A.

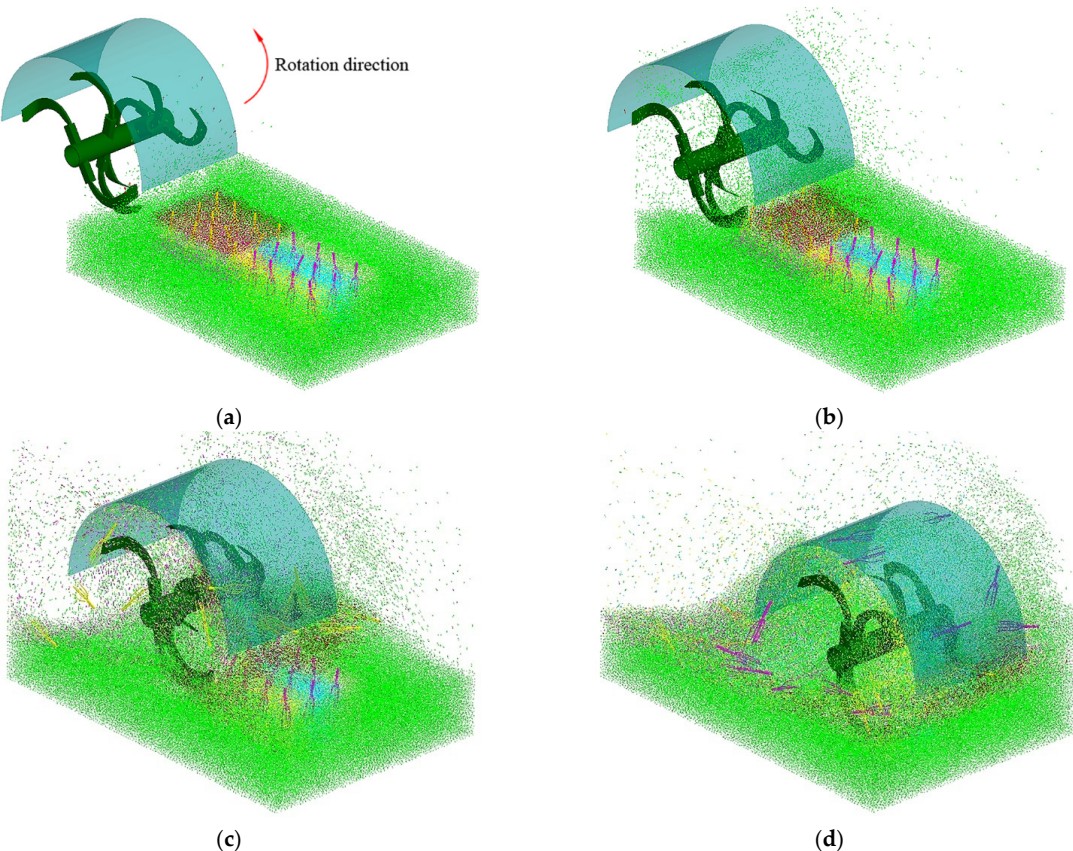

**Figure 12.** Counter-rotation *Cyperus esculentus* harvesting operation. (**a**) 0.5 s simulation moment; (**b**) 1 s simulation moment; (**c**) 2 s simulation moment; and (**d**) 3 s simulation moment.

The degree of fragmentation of *Cyperus esculentus* agglomerates is an important index to evaluate the harvesting performance of *Cyperus esculentus*. Since the movement of *Cyperus esculentus* roots system and soil is caused by the movement of the rotary tillage blade, the movement speed of both can reflect the operation situation in areas A and B from the side. This paper uses the speed of *Cyperus esculentus* and soil in areas A and B to indicate the degree of fragmentation of *Cyperus esculentus* agglomerates in order to further analyze the movement mechanism of soil and *Cyperus esculentus* roots in two areas, A and B, under the action of a rotary tillage blade from the microscopic perspective. This paper focuses on the soil and *Cyperus esculentus* roots disturbance law in areas A and B under counter-rotation operation, as shown in Figure 13.

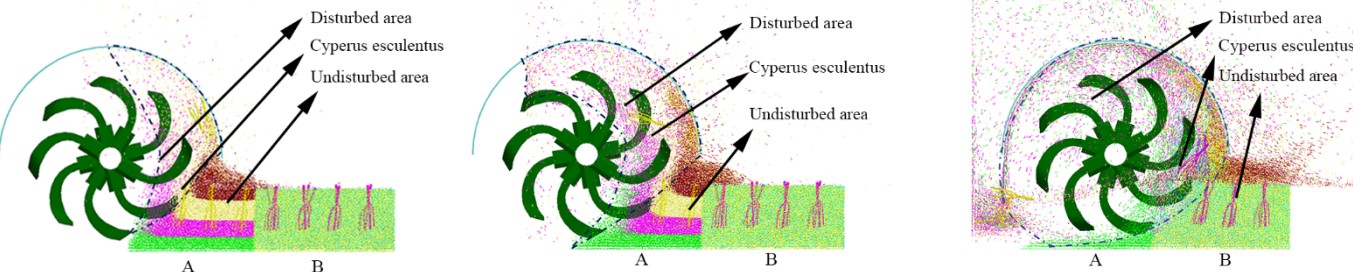

**Figure 13.** Analysis of *Cyperus esculentus*-soil movement behavior in area A and B under counter-rotation operation.

In area A, the rotary tillage blade first contacts the soil in Vertical Section 4 and Vertical Section 3, cutting the soil from bottom to top. The degree of *Cyperus esculentus* agglomerate disturbance gradually increased under the action of the rotary tillage blade, and the soil and *Cyperus esculentus* roots began to be thrown along the tangential direction of the retaining plate as the rotary tillage blade kept cutting the soil. As the simulation test proceeded, the soil disturbance in Vertical Section 2 and Vertical Section 1 gradually increased, and the soil and *Cyperus esculentus* disturbance in Vertical Section 2 and Vertical Section 3 were better than the other two areas from the overall perspective. In area B, the rotary tillage blade acted on three areas at the same time, and the *Cyperus esculentus* roots and soil gradually moved along the tangential direction of the retaining plate with the continuous cutting of the rotary tillage blade. As the simulation experiment continued, the difference in soil disturbance in Horizontal Section 1, Horizontal Section 2, and Horizontal Section 3 was not noticeable.

### 3.1.2. The Analysis of *Cyperus esculentus*–Soil Movement Behavior in Positive Operation

The movement pattern of the *Cyperus esculentus* agglomerates under the positive digging device is shown in Figure 14. As shown in Figure 14a, when the rotary tillage blade first touches the soil, the soil in the upper layer of the *Cyperus esculentus* agglomerate starts to disturb, and with the compound movement of the rotary tillage blade, the layers of the soil produce a downward movement. As shown in Figure 14b, with the simulation test, the rotary tillage blade began to contact the *Cyperus esculentus* agglomerates in area A. At this time, the rotary tillage blade moved from top to bottom, from near to far, in turn, forming the *Cyperus esculentus* digging operation. As shown in Figure 14c, with the compound movement of the rotary tillage blade, the *Cyperus esculentus* roots system, and soil under different soil layers moved backward continuously and in an orderly way along the tangential direction of the blade tip under the action of the rotary tillage blade. As shown in Figure 14d, when the rotary tillage blade operated in area B, the disturbance process and movement pattern of *Cyperus esculentus* roots system and soil was the same as that in area A.

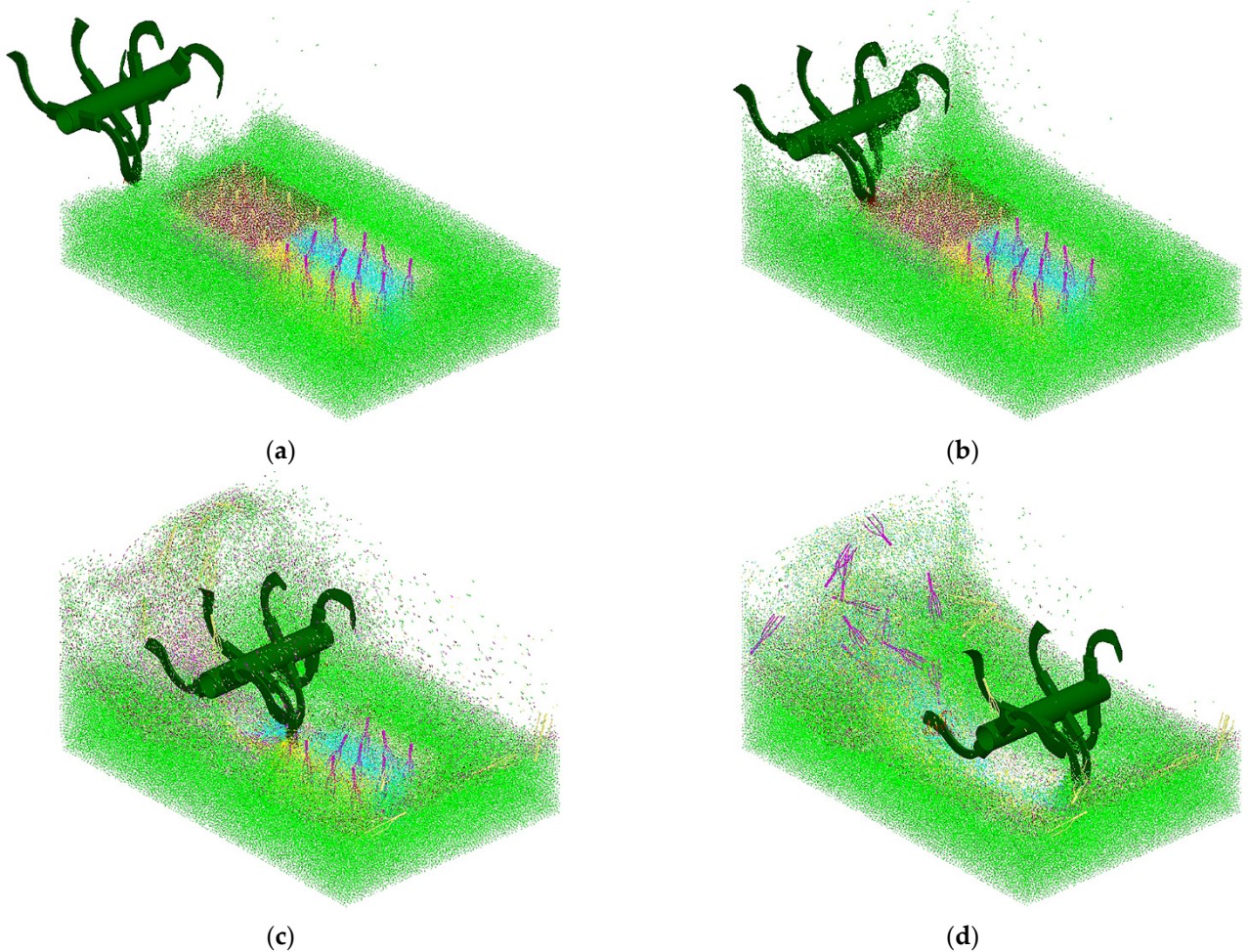

**Figure 14.** Positive rotation *Cyperus esculentus* harvesting operation. (**a**) 0.5 s simulation moment; (**b**) 1 s simulation moment; (**c**) 2 s simulation moment; and (**d**) 3 s simulation moment.

In order to further analyze the movement mechanism of soil and *Cyperus esculentus* roots in two areas, A and B, under the action of a rotary tillage blade from the microscopic perspective, this paper focuses on the soil and *Cyperus esculentus* roots disturbance law in areas A and B under positive rotation operation, as shown in Figure 15.

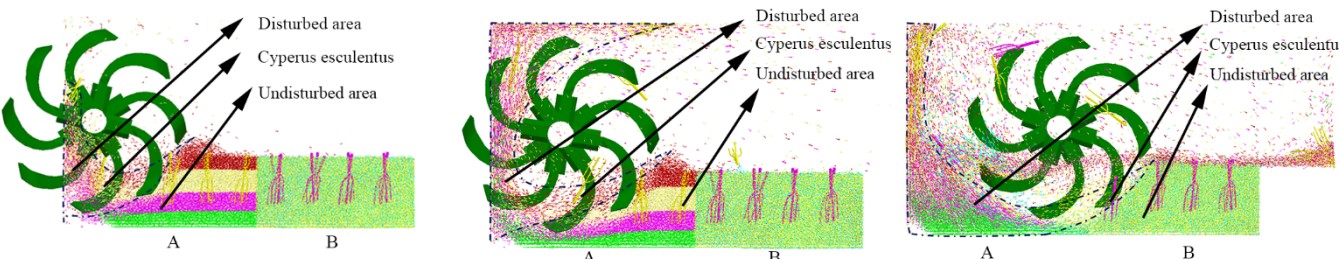

**Figure 15.** Analysis of *Cyperus esculentus*-soil movement behavior in area A and B under positive operation.

In area A, the rotary tillage blade first touches the soil in Vertical Section 1 and Vertical Section 2, which protrudes along the tangential direction of the blade tip under the action of the rotary tillage blade. With the continuous cutting of the rotary tillage blade, the soil and the root system of *Cyperus esculentus* start to move along the tangential di-rection of the blade tip. As the simulation experiment proceeded, the soil disturbance in Vertical Section 3 and Vertical Section 4 gradually increased, and the soil disturbance

and fragmentation of *Cyperus esculentus* aggregated in Vertical Section 1 and Vertical Section 2 were better than the other two areas in general. In area B, the rotary tillage blade acted on three areas at the same time, and the *Cyperus esculentus* roots and soil gradually moved along the tangential direction of the blade tip as the rotary tillage blade kept cutting. As the simulation experiment continued, the difference in soil disturbance in the three areas was not noticeable.

### 3.2. The Analysis of Cyperus esculentus–Soil Disturbance under Positive and Counter-Rotation Operations

3.2.1. The Analysis of Soil Disturbance under Positive and Counter-Rotation Operations

In order to analyze the soil disturbance pattern in area A under positive and counter-rotation operations, this paper focuses on the movement of soil within 0.5~3 s, as shown in Figure 16. The soil disturbance in the counter-rotation operation is shown in Figure 16a. Between 0.5~0.8 s, the soil disturbance of each layer was close to a stationary state. Between 0.8~2 s, the disturbance degree of each layer began to show a significant change. In 2~3 s, the disturbance degree of each layer of soil gradually decreased, and at 3 s, the average speed of each layer of soil was 0.25 m/s. The order of soil disturbance in a vertical direction of counter-rotation operation was Vertical Section 3 > Vertical Section 2 > Vertical Section 1 > Vertical Section 4, which verifies the conclusions of the previous analysis of soil movement.

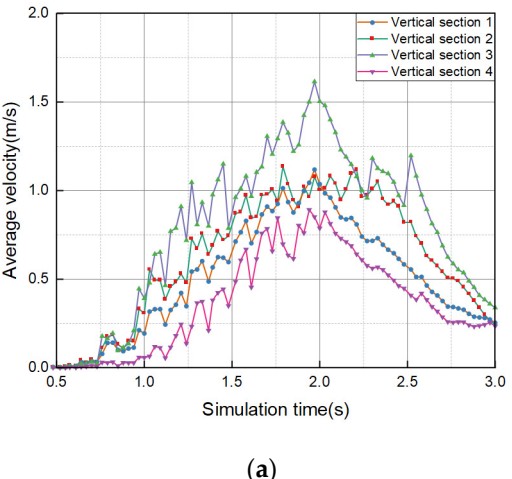

(**a**)

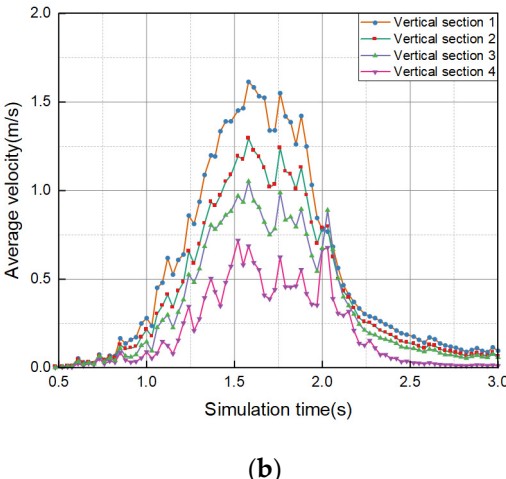

(**b**)

**Figure 16.** Soil disturbance in area A in positive and counter-rotation operation. (**a**) Soil disturbance in counter-rotation operation; (**b**) soil disturbance in positive operation.

The soil disturbance in the positive rotation operation is shown in Figure 16b. Between 0.5 and 0.8 s, the soil disturbance in each layer gradually increased, which was caused by the rotary tillage blade just contacting the soil. Between 0.8 and 1.6 s, with the rotary tillage blade cutting continuously, the soil disturbance in each layer began to change significantly, and at 1.6 s, the soil disturbance in each layer reached the maximum. Between 1.6 and 2 s, the soil disturbance of each layer reached a relatively stable state. Between 2 and 3 s, the soil disturbance of each layer gradually decreased and finally tended towards a stationary state. The sequence of soil disturbance in a vertical direction of positive rotation operation was Vertical Section 1 > Vertical Section 2 > Vertical Section 3 > Vertical Section 4, which verifies the conclusions of the previous analysis of soil movement.

Compared with the analysis of soil disturbance in area A by positive rotation and counter-rotation operation, the disturbance intensity and effective disturbance time of soil by counter-rotation operation are better than that of positive rotation operation.

In order to compare and analyze the soil disturbance law in area B under positive and counter-rotation, this paper focuses on the movement of soil in 1.5~4 s, as shown in

Figure 17. The soil disturbance in the counter-rotation operation is shown in Figure 17a. Between 1.5~1.6 s, the rotary tillage blade had not yet touched the soil in area B, and the soil was in a stationary state. Between 1.6 and 3 s, as the rotary tillage blade kept cutting the soil, the disturbance of soil in each layer of area B gradually increased and reached the maximum at about 2.8 s. Between 3 and 4 s, the movement speed of soil in each layer gradually decreased and finally tended to 0.25 m/s. The soil disturbance in the positive rotation operation is shown in Figure 17b. Between 1.5~1.8 s, the rotary tillage blade had not yet touched the soil in area B, and the soil was in a stationary state. Between 1.8 and 3 s, as the rotary tillage blade kept cutting the soil, the soil in each layer of the B area was disturbed gradually under the action of the rotary tillage blade and reached the maximum disturbance at about 2.7 s. Between 3 and 4 s, the movement speed of the soil in each layer gradually decreased and finally reached a stationary state. Since the rotary tillage blade acted on the B area at the same time, the difference in soil disturbance in Horizontal Section 1, Horizontal Section 2, and Horizontal Section 3 was not noticeable.

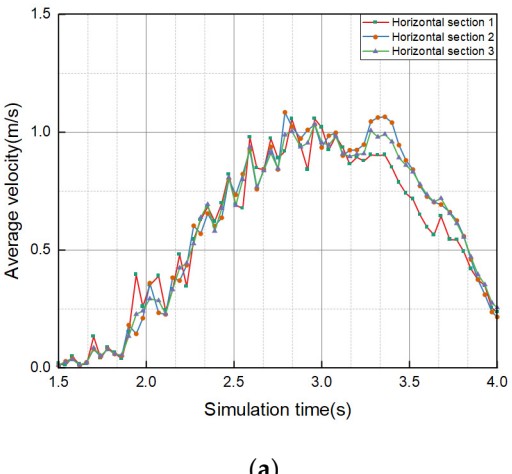
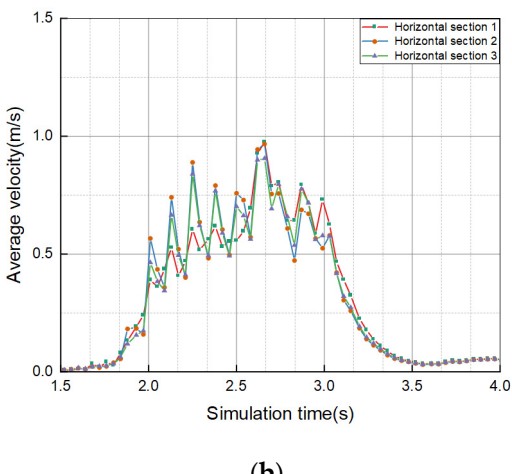

(**a**)　　　　　　　　　　　　　　　　　　(**b**)

**Figure 17.** Soil disturbance in area B in positive and counter-rotation operation. (**a**) Soil disturbance in counter-rotation operation; (**b**) soil disturbance in positive operation.

Comparing the disturbance of soil in area B by positive rotation and counter-rotation operation, the disturbance of soil by counter-rotation operation was better than that by positive rotation. Compared with the positive rotation operation, the counter-rotation operation for soil disturbance intensity and effective disturbance time increased by 166.67% and 133.33%, respectively.

3.2.2. The Analysis of Soil Disturbance under Positive and Counter-Rotation Operations

In order to compare and analyze the root movement pattern of *Cyperus esculentus* in area A under positive and counter-rotation operation, this paper focuses on the root movement of *Cyperus esculentus* in 0.5~3 s, as shown in Figure 18. The disturbance of the *Cyperus esculentus* in counter-rotation operation is shown in Figure 18a. Between 0.5 and 1 s, the rotary tillage blade did not touch the *Cyperus esculentus*, but the movement trend was generated under the disturbance of the soil. Between 1 and 2 s, the soil in each area was disturbed to different degrees under the cutting of the rotary tillage blade. Moreover, the *Cyperus esculentus* started to move. At the moment of 1.8 s, the movement speed of the *Cyperus esculentus* in each row reached the maximum, which is consistent with the counter-rotation action. At 1.8 s, the root velocity of *Cyperus esculentus* reached the maximum, which was consistent with the maximum velocity of soil movement in each layer under the counter-rotation. Between 2 and 3 s, the velocity of *Cyperus esculentus* was constant because the *Cyperus esculentus* was being thrown along the tangent line of the retaining plate under the action of the rotary tillage blade and soil.

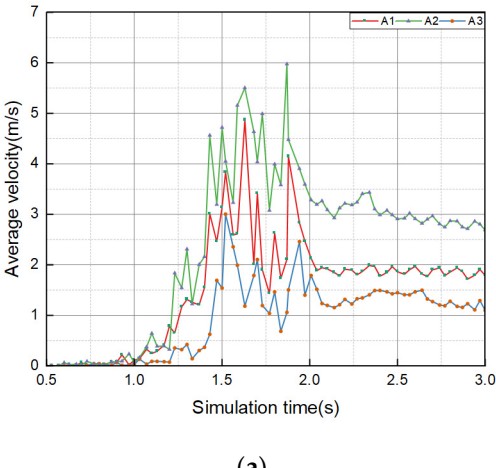
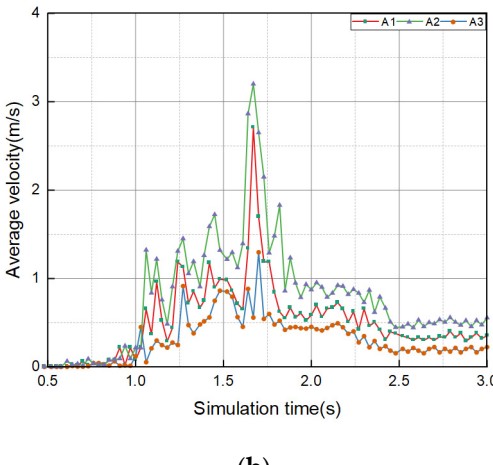

(**a**)                                         (**b**)

**Figure 18.** *Cyperus esculentus* disturbance in area A in positive and counter-rotation operation. (**a**) *Cyperus esculentus* disturbance in counter-rotation operation; (**b**) *Cyperus esculentus* disturbance in positive operation.

The disturbance of the *Cyperus esculentus* in positive rotation operation is shown in Figure 18b. Between 0.5~1 s, the rotary tillage blade was cutting soil but did not touch the *Cyperus esculentus*, and the *Cyperus esculentus* produced a movement trend under the disturbance of the soil. Between 1 and 2 s, with the reciprocal movement of the rotary tillage blade, each layer of soil was disturbed to different degrees under the rotary tillage blade cutting, and the root system of *Cyperus esculentus* started to move because the root system of *Cyperus esculentus* in row A2 received two groups of rotary cutter disturbance at the same time, resulting in its movement speed being higher than the root system of *Cyperus esculentus* in groups A1 and A3. At 1.6 s, the motion speed of the *Cyperus esculentus* in each row reached the maximum, which was consistent with the moment when the movement speed of each soil layer reached the maximum. Between 2 and 3 s, the movement speed of the *Cyperus esculentus* gradually decreased because the soil disturbance ability of the rotary tillage blade gradually decreased. As the experiment proceeded, the root velocities of all the roots of the *Cyperus esculentus* eventually tended to 0 m/s.

Comparing the positive rotation and counter-rotation operation on the root movement of *Cyperus esculentus* in area A, the disturbance intensity and effective disturbance time of the counter-rotation operation were better than those of the positive rotation operation.

In order to compare and analyze the root movement pattern of *Cyperus esculentus* in area B under positive rotation and counter-rotation operation, this paper focuses on the root movement of *Cyperus esculentus* in 1.5~4 s, as shown in Figure 19.

The disturbance of the *Cyperus esculentus* in counter-rotation operation is shown in Figure 19a. Between 1.5~2 s, the movement trend of the *Cyperus esculentus* is consistent with that of positive rotation excavation. Between 2~3 s, the motion velocity of the *Cyperus esculentus* in each region gradually increased under the action of soil motion, and the motion velocity of the *Cyperus esculentus* reached the maximum at 2.8 s. Between 3~4 s, the *Cyperus esculentus* in each region made a throwing motion along the tangent line of the retaining plate, and the motion velocity reached a steady state.

The root disturbance of *Cyperus esculentus* in positive rotation operation is shown in Figure 19b. Between 1.5~2 s, the root system of *Cyperus esculentus* gradually produced the movement trend under soil disturbance. Moreover, between 2~3.3 s, with the compound movement of the rotary tillage blade, the degree of soil disturbance in each area gradually increased, and the movement speed of *Cyperus esculentus* gradually increased under the action of soil movement, and the movement speed of *Cyperus esculentus* reached the maximum at 2.7 s. Due to the root system of *Cyperus esculentus*, row B2 received two groups of rotary tillage blade disturbance at the same time, which led to its movement speed being

higher than that of the B1 and B3 groups of *Cyperus esculentus*. Between 3.3 and 4 s, the speed of the *Cyperus esculentus* decreased due to the decrease of the rotary tillage blade's ability to disturb the soil and finally reached a stationary state.

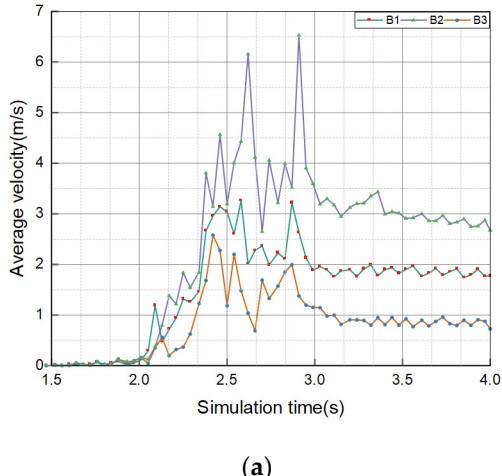

(**a**)

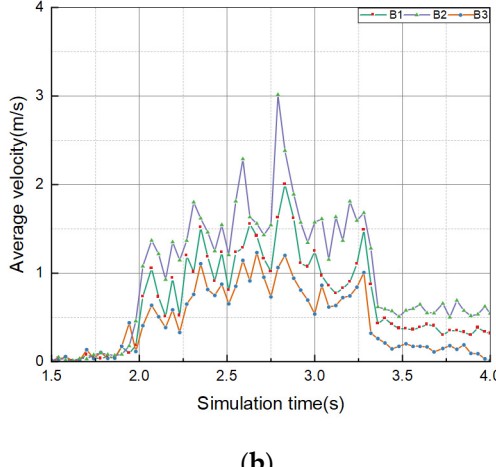

(**b**)

**Figure 19.** *Cyperus esculentus* disturbance in area B in positive and counter-rotation operation. (**a**) *Cyperus esculentus* disturbance in counter-rotation operation; (**b**) *Cyperus esculentus* disturbance in positive operation.

Comparing the positive and counter-rotation operation on the root movement of *Cyperus esculentus* in area B, the disturbance of *Cyperus esculentus* by counter-rotation operation was better than that by positive rotation. Compared with the forward rotation operation, the intensity and effective disturbance time of counter-rotation operation on the *Cyperus esculentus* increased by 297.78% and 133.33%, respectively.

### 3.3. The Effect of Different Operating Parameters on the Intensity of Cyperus esculentus–Soil Disturbance

3.3.1. The Effect of Rotation Speed of Rotary Tillage Blade on the Intensity of *Cyperus esculentus*–Soil Disturbance

In order to analyze the dynamic relationship between the rotary tillage blade and the soil and *Cyperus esculentus* under different working parameters, this paper systematically studied the influence of rotation speed of the rotary tillage blade, forward speed, and working depth on the movement characteristics of the soil and *Cyperus esculentus*. The specific working parameters are shown in Table 3.

**Table 3.** The working conditions in the simulation.

| No. | Rotary Speed (rpm) | Forward Speed (m/s) | Working Depth (mm) |
|-----|--------------------|---------------------|--------------------|
| 1 | 270 | 0.5 | 150 |
| 2 | 300 | 0.5 | 150 |
| 3 | 330 | 0.5 | 150 |
| 4 | 270 | 0.5 | 150 |
| 5 | 270 | 0.7 | 150 |
| 6 | 270 | 0.9 | 150 |
| 7 | 270 | 0.5 | 150 |
| 8 | 270 | 0.5 | 160 |
| 9 | 270 | 0.5 | 170 |

Tests 1, 2, and 3 in the table of working parameters were conducted to compare the rotation speed of the rotary tillage blade on the movement characteristics of the soil and root system of *Cyperus esculentus*. The disturbance pattern of rotation speed of the

rotary tillage blade on soil and *Cyperus esculentus* in area A is shown in Figure 20. The disturbance pattern of rotation speed of the rotary tillage blade on soil in area A is shown in Figure 20a. In the vertical direction, the degree of disturbance of soil and *Cyperus esculentus* by positive and counter-rotation operation increases with the increase of the rotation speed of the rotary tillage blade. For the positive rotation operation, with the increase of rotation speed of the rotary tillage blade, the soil disturbance intensity increased by 33.33%, 28.89%, and 21.62% in Section 1, Section 2, and Section 3, respectively, and the soil disturbance intensity increased significantly to 91.67% in the Section 4 area. For the counter-rotation operation, the soil disturbance intensity increased by 21.55%, 26.77%, and 32.35% for Section 1, Section 2, and Section 3, respectively, with the increase of rotation speed of the rotary tillage blade, and the soil disturbance intensity increased significantly to 123.53% in the Section 4 area. The disturbance pattern of rotation speed of the rotary tillage blade on *Cyperus esculentus* in area A is shown in Figure 20b. The most significant effect of the rotation speed of the rotary tillage blade variation on the root disturbance intensity of *Cyperus esculentus* was in row A2. The disturbance intensity of positive rotation operation on the root system of *Cyperus esculentus* increased by 28.85% at A2 and 35.29% at A2 by counter-rotation operation.

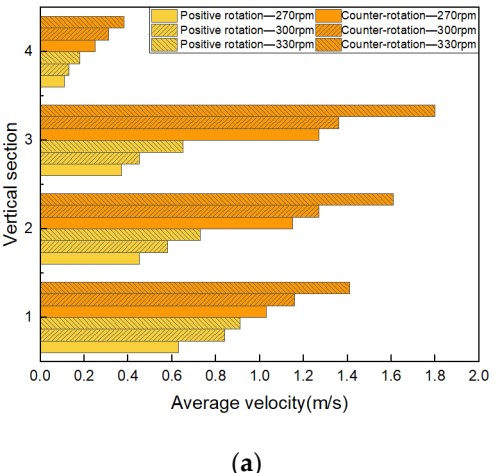

(a)

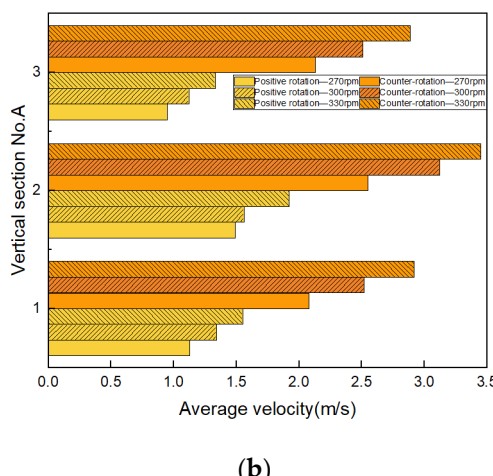

(b)

**Figure 20.** Effect of the rotation speed of the rotary tillage blade on *Cyperus esculentus*-soil disturbance in area A. (**a**) Soil disturbance in area A; (**b**) *Cyperus esculentus* disturbance in area A.

The disturbance pattern of rotation speed of the rotary tillage blade on soil and *Cyperus esculentus* in area B is shown in Figure 21. In the horizontal direction, the degree of disturbance of soil and *Cyperus esculentus* by positive and counter-rotation increases with the increase of rotation speed of the rotary tillage blade. The soil disturbance pattern of rotation speed of the rotary tillage blade in area B is shown in Figure 21a. For positive rotation operation, with the increase of rotation speed of the rotary tillage blade, Section 1, Section 2, and Section 3 areas increased by 12.07%, 17.18%, and 12.28%, respectively. For counter-rotation operation, Section 1, Section 2, and Section 3 areas increased by 13.82%, 17.42%, and 14.40%, respectively. The pattern of rotation speed of the rotary tillage blade on root disturbance of *Cyperus esculentus* in area B is shown in Figure 21b. The most significant effect of the rotation speed of the rotary tillage blade variation on root disturbance intensity of *Cyperus esculentus* was in row B2. There was a 27.40% and 29.06% increase in root disturbance intensity at B2 by positive and counter-rotation operation, respectively.

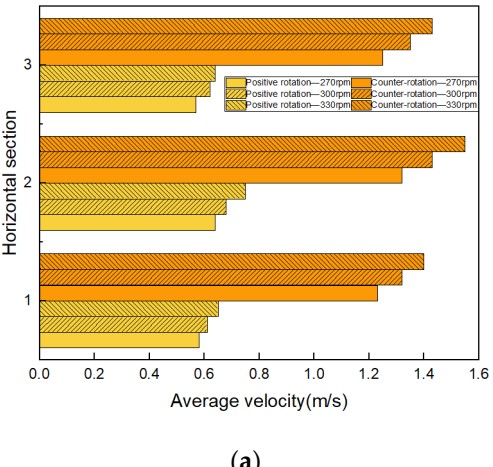
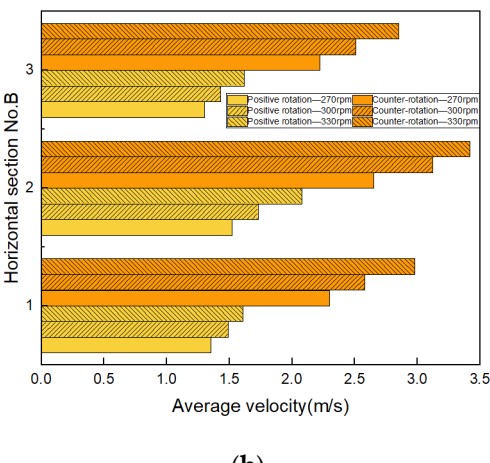

**(a)**                      **(b)**

**Figure 21.** Effect of the rotation speed of the rotary tillage blade on *Cyperus esculentus*–soil disturbance in area B. (**a**) Soil disturbance in area B; (**b**) *Cyperus esculentus* disturbance in area B.

The comparative analysis of rotation speed of the rotary tillage blade on the root and soil disturbance patterns of *Cyperus esculentus* in areas A and B showed that the increase in rotation speed of the rotary tillage blade improved the harvesting efficiency of *Cyperus esculentus* to some extent.

3.3.2. The Effect of Forward Speed on the Intensity of *Cyperus esculentus*–Soil Disturbance

Tests 4, 5, and 6 in the table of working parameters were conducted in order to compare the forward speed on the movement characteristics of soil and root system of *Cyperus esculentus*; the disturbance pattern of forward speed on soil and *Cyperus esculentus* in area A is shown in Figure 22. The soil disturbance pattern of the rotary tillage blade forward speed on area A is shown in Figure 22a. With the increase in forward speed, the degree of disturbance of soil by positive rotation operation gradually decreased, and the soil disturbance intensity decreased by 12.31% at Section 1. With the increase in forward speed, the degree of soil disturbance by counter-rotating operation only increased by 5.92% in Section 3. The disturbance pattern of rotary tillage blade forward speed on the *Cyperus esculentus* in area A is shown in Figure 22b. The disturbance intensity of the *Cyperus esculentus* gradually decreased with the increase of forward speed in the positive rotation operation and only increased by 7.83% in the counter-rotation operation at A2.

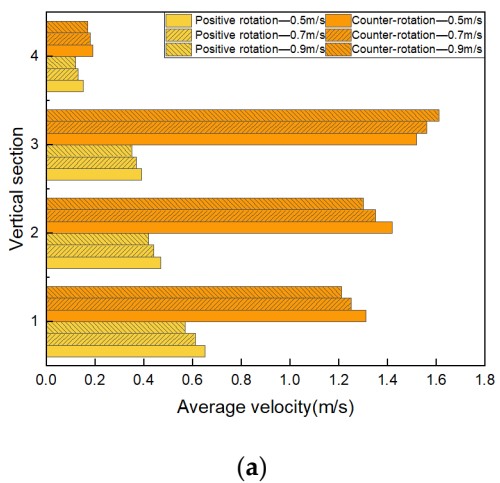
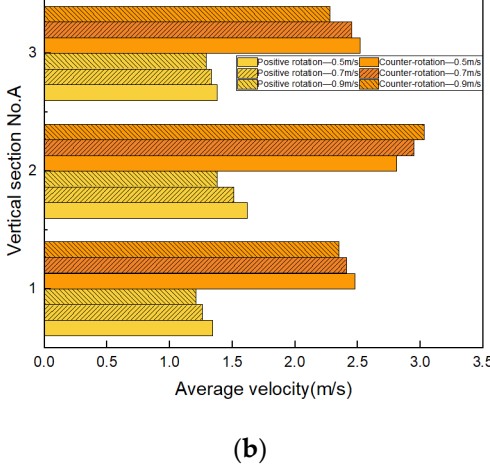

**(a)**                      **(b)**

**Figure 22.** Effect of the forward speed on *Cyperus esculentus*-soil disturbance in area A. (**a**) Soil disturbance in area A; (**b**) *Cyperus esculentus* disturbance in area A.



The disturbance pattern of forward speed on soil and *Cyperus esculentus* in area B is shown in Figure 23. The soil disturbance pattern of the rotary tillage blade forward speed on area B is shown in Figure 23a, the intensity of soil disturbance in Section 2 and Section 3 was reduced by 12.07% and 12.59% by positive and counter-rotating operations, respectively. The disturbance pattern of rotary tillage blade forward speed on the *Cyperus esculentus* in area B is shown in Figure 23b. The intensity of root disturbance of *Cyperus esculentus* in Section 2 and Section 3 was reduced by 16.13% and 7.98% for positive and counter-rotation operations, respectively.

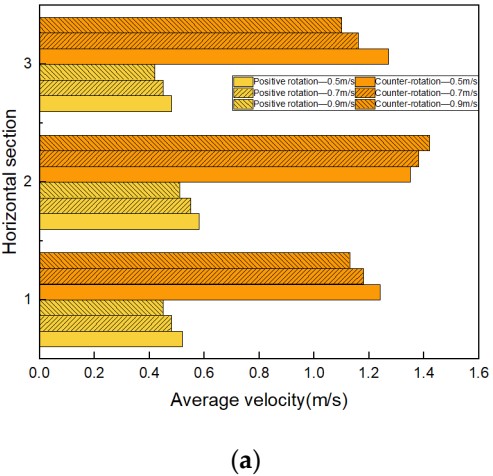 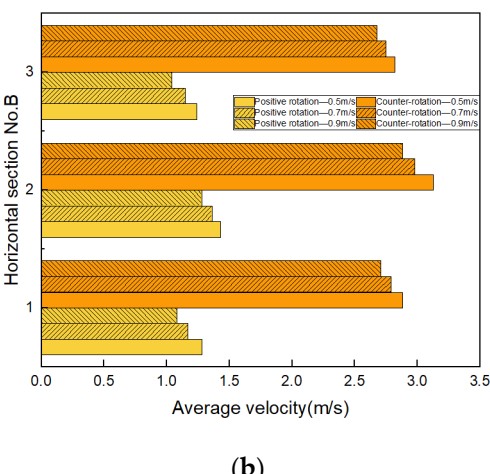

**(a)**             **(b)**

**Figure 23.** Effect of the forward speed on *Cyperus esculentus*–soil disturbance in area B. (**a**) Soil disturbance in area B; (**b**) *Cyperus esculentus* disturbance in area B.

The comparative analysis of the forward speed on the *Cyperus esculentus* and soil disturbance pattern of *Cyperus esculentus* in areas A and B showed that the increase in forward speed reduced the harvesting efficiency of *Cyperus esculentus* to some extent.

### 3.3.3. The Effect of Working Depth on the Intensity of *Cyperus esculentus*–Soil Disturbance

Tests 7, 8, and 9 in the table of working parameters were conducted in order to compare the working depth for the study of the movement characteristics of *Cyperus esculentus* agglomerates; the disturbance pattern of working depth on soil and *Cyperus esculentus* in area A is shown in Figure 24. The soil disturbance pattern of the rotary tillage blade working depth in area A is shown in Figure 24a. With the increase of working depth, the intensity of soil disturbance in all layers of positive rotation excavation gradually increased by 20.97%, 23.40%, and 23.68% in Section 1, Section 2, and Section 3, respectively, and sharply increased by 92.86% in Section 4. The counter-rotating operation increased by 9.72% and 106.67% in Section 3 and Section 4, respectively, although the intensity of soil disturbance decreased in Section 1 and Section 2.

The disturbance pattern of rotary tillage blade working depth on the *Cyperus esculentus* in area A is shown in Figure 24b. With the increase of working depth, the disturbance intensity of positive and counter-rotating operation on the root system of *Cyperus esculentus* in area A gradually increased and was most significant at A2, which increased by 18.18% and 10.92%, respectively.

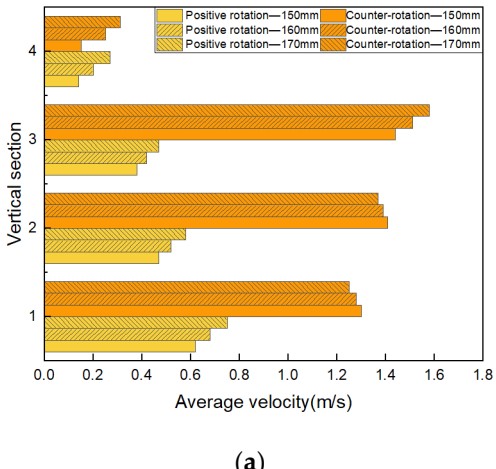

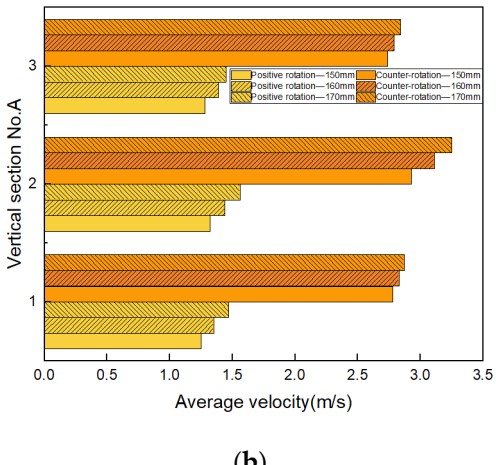

(**a**)                          (**b**)

**Figure 24.** Effect of the working depth on *Cyperus esculentus*-soil disturbance in area A. (**a**) Soil disturbance in area A; (**b**) *Cyperus esculentus* disturbance in area A.

The disturbance pattern of working depth on soil and *Cyperus esculentus* in area B is shown in Figure 25. The pattern of soil disturbance in area B at the working depth of the rotary tillage blade is shown in Figure 25a. With the increase in working depth, the intensity of soil disturbance in all layers of the positive rotary operation gradually increased, with the highest increase of 17.91% in Section 3. The intensity of soil disturbance in Section 1 and Section 3 decreased with the increase in the working depth of counter-rotating operation but increased by 21.37% in Section 2. The disturbance pattern of rotary tillage blade operating depth on the root system of *Cyperus esculentus* in area B is shown in Figure 25b. With the increase of working depth, the disturbance intensity of positive and counter-rotating operation on the root system of *Cyperus esculentus* in area B gradually increased.

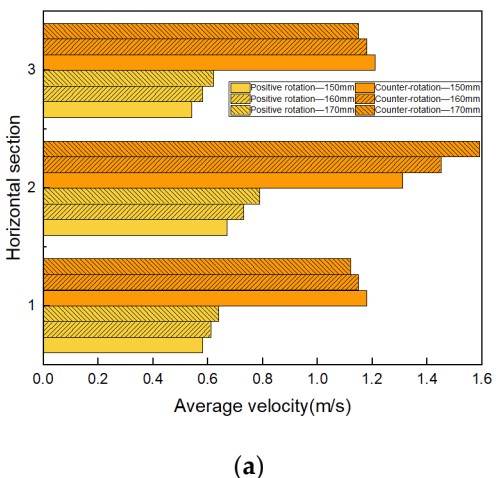

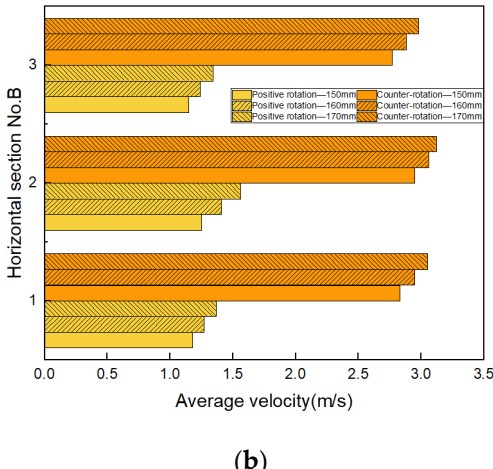

(**a**)                          (**b**)

**Figure 25.** Effect of the working depth on *Cyperus esculentus*-soil disturbance in area B. (**a**) Soil disturbance in area; (**b**) *Cyperus esculentus* disturbance in area B.

The comparative analysis of the working depth on the *Cyperus esculentus* and soil disturbance pattern of *Cyperus esculentus* in areas A and B showed that the increase in working depth improved the harvesting efficiency of *Cyperus esculentus* to a certain extent.

*3.4. The Results of the Field Test*

The purpose of the field test was to verify whether the *Cyperus esculentus* counter-rotation digging device had the efficient operation, and the test results are shown in Table 4.

The test results showed that the maximum value of soil fragmentation rate was 94.27%, the minimum value was 92.48%, the average value was 93.50%, and the maximum value of *Cyperus esculentus* damage rate was 1.52%, the minimum value was 1.35%, and the average value was 1.42%, and the maximum value of *Cyperus esculentus* buried fruit rate was 1.51%, the minimum value was 1.28%, and the average value was 1.38% during the operation of the counter-rotation digging device.

**Table 4.** The results of the comparative field test of *Cyperus esculentus*.

| No. | Counter-Rotating Excavation | | | Positive-Rotating Excavation | | |
| --- | --- | --- | --- | --- | --- | --- |
| | Soil Fragmentation Rate/% | Damage Rate/% | Buried Fruit Rate/% | Soil Fragmentation Rate/% | Damage Rate/% | Buried Fruit Rate/% |
| 1 | 93.56 | 1.35 | 1.28 | 90.67 | 1.56 | 1.37 |
| 2 | 92.48 | 1.41 | 1.36 | 89.96 | 1.54 | 1.43 |
| 3 | 93.74 | 1.45 | 1.33 | 88.57 | 1.61 | 1.45 |
| 4 | 93.45 | 1.38 | 1.42 | 90.28 | 1.57 | 1.56 |
| 5 | 94.27 | 1.52 | 1.51 | 89.67 | 1.68 | 1.64 |
| Average value | 93.50 | 1.42 | 1.38 | 89.83 | 1.59 | 1.49 |

The maximum value of the soil fragmentation rate was 90.67%, the minimum value was 88.57%, the average value was 89.83%, and the maximum value of the *Cyperus esculentus* damage rate was 1.68%, the minimum value was 1.54%, and the average value was 1.59%, and the maximum value of the *Cyperus esculentus* buried fruit rate was 1.64%, the minimum value was 1.37%, and the average value was 1.49% during the operation of the positive rotation excavation device. Compared with the positive rotation operation, the *Cyperus esculentus* counter-rotation soil fragmentation rate increased by 4.09%, the *Cyperus esculentus* damage rate increased by 10.69%, and the buried fruit rate increased by 7.38%.

Compared with the positive rotation operation, the *Cyperus esculentus* counter-rotation harvesting operation can improve the soil cutting speed and operational stability and realize the parabolic movement of *Cyperus esculentus*–soil along the tangential direction of the soil guide plate. Meanwhile, it can enhance the degree of disturbance between *Cyperus esculentus* and soil, ensure the efficiency of *Cyperus esculentus* agglomerate fragmentation and *Cyperus esculentus* harvesting efficiency, and meet the basic requirements of *Cyperus esculentus* harvesting.

## 4. Discussion

At present, many scholars have conducted in-depth research on the mechanized harvesting mechanism of *Cyperus esculentus* to improve the harvesting efficiency of *Cyperus esculentus* further. Zhu et al. [2] accurately simulated the breakage of the *Cyperus esculentus* plant and the rotating tiller and the plant–soil–rotating tiller discrete element model was constructed to conduct simulation tests with power consumption and plant breakage rate as evaluation indicators. The results showed that the average power consumption of the IT245P rotary tillage knife was reduced by 13.10%, and the plant breakage rate was increased by 11.75% compared to the IT245. He et al. [6,12] solved the problems of high working resistance, low soil breaking rate, and high buried fruit rate in the harvesting process of *Cyperus esculentus* and a new rotary tillage knife was designed by combining EDEM discrete element simulation tests. The results showed that, compared with the standard rotary tillage knife, the new rotary tillage knife reduced the buried fruit rate of *Cyperus esculentus* by 1.2% and increased the soil breaking rate by 1.3%, thus improving the harvesting efficiency of *Cyperus esculentus*.

This paper proposed a method of *Cyperus esculentus* harvesting based on counter-rotation digging. The mechanism of interaction between the rotary tillage blade and *Cyperus esculentus*–soil was systematically investigated, and the vertical and horizontal disturbance performance of the positive and counter-rotating harvesting methods on soil

and *Cyperus esculentus* was compared and analyzed. Compared with the positive rotation operation, the *Cyperus esculentus* counter-rotation soil fragmentation rate increased by 4.09%, the *Cyperus esculentus* damage rate decreased by 10.69%, and the buried fruit rate decreased by 7.38%. To further verify the advanced state of the test results, combined with related research [2,6,12,35], a comparative analysis with related studies was carried out in this paper, and the test results are shown in Table 5. This indicates that the design of the counter-rotating digging device can further improve the harvesting efficiency of *Cyperus esculentus*.

**Table 5.** The results of comparisons and discussions in the field test.

| Type of Excavators | Soil Fragmentation Rate/% | Damage Rate/% | Buried Fruit Rate/% |
| --- | --- | --- | --- |
| Counter-rotating excavation | 93.50 | 1.42 | 1.38 |
| New rotary tillage knife | 92.18 | / | 2.07 |
| IT245P | 91.73 | 1.75 | / |

**5. Conclusions**

(1) To study the dynamic relationship between *Cyperus esculentus*–soil and rotary tillage blade under positive and counter-rotation operation, a discrete element model of *Cyperus esculentus*–soil–rotary tillage blade was established in this paper, and the movement behaviors of soil and *Cyperus esculentus* were analyzed. The experiments showed that in the AB area, the disturbance intensity of soil and *Cyperus esculentus* increased by 166.67% and 297.78% for counter-rotation operation compared with the positive rotation operation, and the effective disturbance time for *Cyperus esculentus* and soil increased by 133.33% compared with the positive rotation operation, indicating that counter-rotation operation increased the harvesting efficiency of *Cyperus esculentus* to some extent.

(2) This paper systematically analyzed the disturbance law of rotary tillage blade rotation speed, forward speed, and working depth on the root system and soil of *Cyperus esculentus*. The test showed that the working depth was the most significant for soil disturbance intensity, with the working depth increasing from 150 mm to 170 mm, the soil disturbance intensity increased by 17.91% and 21.37% for positive and counter-rotation rotation, respectively. The rotation speed of the rotary tillage blade was the most significant for the disturbance intensity of the root system of the *Cyperus esculentus*. With the increase of the rotary tillage blade rotation speed from 270 rpm to 330 rpm, the disturbance intensity of the root system of the *Cyperus esculentus* increased by 28.85% and 35.29% for the positive and counter-rotation operations, respectively.

(3) This paper conducted a comparative field test of *Cyperus esculentus* harvesting. It systematically analyzed the field test results using soil fragmentation rate and *Cyperus esculentus* damage rate as the test indexes. The results showed that compared with the positive rotation operation, the *Cyperus esculentus* counter-rotation soil fragmentation rate increased by 4.09%, the *Cyperus esculentus* damage rate decreased by 10.69%, and the buried fruit rate decreased by 7.38%.

**Author Contributions:** Conceptualization, Z.Z. and D.W.; methodology, Z.Z.; software, Z.Z.; validation, Z.Z.; formal analysis, Z.Z.; investigation, Z.Z., Z.C. and N.X.; resources, Z.Z. and X.Z.; data curation, Z.Z.; writing—original draft preparation, Z.Z.; writing—review and editing, Z.Z.; visualization, Z.Z. and P.G.; supervision, Z.Z. and Z.G.; project administration, X.H., S.S. and J.H.; funding acquisition, X.H. and D.W. All authors have read and agreed to the published version of the manuscript.

**Funding:** This research was funded by the Autonomous Region Science and Technology Support Project Plan (Grant NO.2020E02112) and Major Science and Technology Projects in Henan Province (Grant NO.211100110100) and Natural Science Foundation of Shandong Province Youth Project (Grant NO. ZR2022QE167).

**Institutional Review Board Statement:** Not applicable.

**Data Availability Statement:** Not applicable.

**Conflicts of Interest:** The authors declare no conflict of interest.

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
