# Peer review of "Analysis of Cyperus esculentus–Soil Dynamic Behavior during Rotary Tillage Based on Discrete Element Method"

_agriculture, doi:10.3390/agriculture13020358_

Round 1

Reviewer 1 Report

Please see the"peer-review-25884958.V1.docx"

Author Response

Please see attached for specific changes to the article

Reviewer 2 Report

(1) In Table 1, the units and values of soil density do not correspond.

(2) How do you check the discrete element model? What is the connection between the simulation and the field test? What is the significance of the simulation?

(3) The parameters analyzed in the field test are too few.

(4) The results and findings can be compared to and discussed in the context of earlier work in the literature.

(5) The unit of time “s” should be lowercase.

(6) Before submitting a revision be sure that your material is grammatically correct,  properly prepared, and formatted. For example: In lines 90-91, “one is the counter-rotation Cyperus esculentus digging device, and one is the positive rotation Cyperus esculentus digging device.” In lines 167-170, “Soil density test using volume V for 100cm3 soil cutting ring blade in 0~50 mm, 50~100mm, 100~150mm, and 150~200mm soil area were repeatedly sampled, using electronic scales for mass measurement, respectively, the total mass m1 and ring knife mass m0, soil density is the measured soil mass to volume ratio.”

Author Response

(The authors gave the same response as above.)

Reviewer 3 Report

1. The first letter of a completed sentence needs to be capitalized. For example, rows 90 and 383. Suggested check throughout.

2. Please explain what the symbol “d1”in Figure 4(a) stands for.

3. What is the basis for the selection of working parameters in line 261 ?

4. For the reader's understanding, we propose to add an arrow indicating the direction of rotation in Figure 12.

5. Whether the rotary tillage equipment has a soil-directed surface in the field trial of positive rotationIf so, then add it accordingly to the simulations in Figure 14, and if not, please unify it with Figure 2.

6. In the discrete element simulation model, the soil particles used in areas A and B are not the same color. Is it the same soil particle? Please explain in the paper.

7. There is an error in subheading 3.3.2. This section describes the effect of forward speed, not RPM.

8. Subheading 3.2.2 is incorrect. Please correct it.

9. When there are many soil particles, it is not easy to analyze the movement pattern of soil vertically and horizontally at the same time, so set up A and B zones. In the analysis of the movement law of Cyperus esculentus, the number of Cyperus esculentus is less, and the distribution position of Cyperus esculentus in A and B area and the contact parameters with soil are exactly the same, under the same operating conditions, the movement law should naturally be the same, why do we have to analyze AB two areas separately? Would it be better to combine the two zones for analysis?

10. When conducting the tests for the operating parameters, three values were set for each factor. However, when conducting the analysis, only the data for the highest and lowest values were listed and compared, and conclusions were drawn, while the data for the intermediate values were not known. For example, the data for the rotary speed of 300rpm is missing. It is recommended that the data corresponding to the intermediate values be completed during the data analysis. Without this part of the data comparison, there would be no point in the existence of a median test. These intermediate values are shown in brackets. (300rpm0.7m/s160mm)

Author Response

(The authors gave the same response as above.)

Round 2

Reviewer 2 Report

The authors gave detailed answers to the questions raised by the reviewers and made good revisions to the manuscript. Thus, in my humble opinion, the manuscript can be accepted. By the way, in Table 1, the units of soil density should be g/cm3.